# Favorable association between early initiation of sodium-glucose cotransporter-2 inhibitors and in-hospital prognosis in acute myocardial infarction

Hung Thanh Quach[1,2☯], Anh Thi Kim Nguyen[1], Bao Quoc Dinh[2], Tuan Minh Nguyen[1,2], Hoang Huy Bui[1], Tuan Ngoc Tran[1], Tien Dung Ho[1], An Le Pham[1,2], Si Van Nguyen [ID][1,2☯]*

1 Department of Cardiology 1, Nguyen Trai Hospital, Ho Chi Minh City, Vietnam, 2 Center for Family Physicians, School of Medicine, University of Medicine and Pharmacy at Ho Chi Minh City, Ho Chi Minh City, Vietnam

☯ These authors contributed equally to this work.
* si.nguyen@ump.edu.vn

## Abstract

### Introduction

Acute myocardial infarction remains a major cause of death and disability worldwide, especially in low- and middle-income countries. Standard early management includes dual antiplatelet therapy, statins, beta-blockers (BB) and renin–angiotensin–aldosterone inhibitors (ACEi/ARB/MRA). Sodium–glucose cotransporter-2 inhibitors (SGLT2i) have been associated with favorable changes in cardiac function in acute myocardial infarction, but their impact on in-hospital mortality has not been well established.

### Objective

To assess the association between early initiation of SGLT2i and in-hospital mortality among patients with acute myocardial infarction.

### Methods

A retrospective, single-center study was conducted on 394 adult patients hospitalized with acute myocardial infarction at Nguyen Trai Hospital, Ho Chi Minh City, between January 2022 and October 2024. The data extraction and analysis were performed from July 31 to October 24, 2024. Patients with incomplete data, secondary acute myocardial infarction, or eGFR < 20 mL/min/1.73 m² were excluded. Data from electronic medical records were analyzed. The primary outcome was in-hospital mortality. Logistic regression was used to identify independent predictors.

**Data availability statement:** Data cannot be shared publicly because of the institutional confidentiality policies. Data are available from the Nguyen Trai Hospital (contact via bv.nguyentrai@bv.nguyentrai.gov.vn.) for researchers who meet the criteria for access to confidential data.

**Funding:** The author(s) received no specific funding for this work.

**Competing interests:** The authors have declared that no competing interests exist.

## Results

Mean age was 66.0 ± 11.7 years; 57.1% were male. SGLT2i was initiated within 24 hours of admission in 23.9% of patients. In-hospital mortality occurred in 53 patients (13.5%). In multivariable analysis, lower left ventricular ejection fraction (OR 0.91, 95% CI: 0.85–0.97; p = 0.003) and sepsis (OR 5.14, 95% CI: 1.04–25.36; p = 0.04) were independently associated with in-hospital mortality. In addition, use of BB/ACEi/ ARB/MRA and early SGLT2i initiation were independently associated with lower in-hospital mortality (OR 0.12, 95% CI: 0.05–0.25; p < 0.001 and OR 0.27, 95% CI: 0.07–0.96; p = 0.04, respectively).

## Conclusions

Together with traditional medical treatment, initiating SGLT2i within 24 hours of admission for acute myocardial infarction was independently associated with lower in-hospital mortality. These findings suggest a potential association between early SGLT2i use and improved in-hospital outcomes and warrant further investigation in prospective randomized studies.

## Introduction

Acute myocardial infarction (AMI) is a leading cause of mortality and disability worldwide, with a particularly severe impact on low- and middle-income countries [1]. Standard pharmacological management following AMI is typically initiated within the first 24 hours and includes dual antiplatelet therapy, statins, beta-blockers, and inhibitors of the renin–angiotensin–aldosterone system [2]. Previous studies have demonstrated that sodium–glucose cotransporter-2 inhibitors (SGLT2i) exerted favorable effects on left ventricular ejection fraction and reduced N-terminal pro B-type natriuretic peptide (NT-proBNP) levels – two important prognostic markers in hospitalized patients with AMI. However, their benefits in reducing hard outcomes such as hospital readmission and mortality have not been established [3,4]. Based on these findings, we hypothesized that the use of SGLT2i might be associated with lower in-hospital mortality in patients following AMI. Therefore, the primary objective of this study was to evaluate the association between SGLT2i use and in-hospital prognosis in patients with AMI.

## Methods

A retrospective study was conducted on 394 adult patients diagnosed with AMI from January 2022 to October 2024 at Nguyen Trai Hospital, Ho Chi Minh City, Vietnam. The data were accessed for research purposes from July 31 to October 24, 2024. The following cases were excluded: (1) inadequate confirmation of AMI diagnosis, (2) incomplete medical records, (3) secondary AMI related to procedures/surgery or type 2 MI, and (4) estimated glomerular filtration rate (eGFR) < 20 mL/min/1.73 m². Baseline characteristics of the study population are presented in Table 1.

**Table 1. Baseline characteristics.**

| | All (N = 394) | In-hospital mortality | | P-value |
| --- | --- | --- | --- | --- |
| | | Event (N = 53) | Non-event (N = 341) | |
| Age (years) | 66.0 ± 11.7 | 71.4 ± 11.5 | 65.3 ± 11.5 | **< 0.001** |
| Male gender | 225 (57.1) | 27 (50.9) | 198 (58.1) | 0.3 |
| Smoking | 37 (9.4) | 2 (3.8) | 35 (10.3) | 0.2 |
| Hypertension | 320 (81.2) | 46 (86.8) | 274 (80.4) | 0.2 |
| Diabetes | 120 (30.5) | 21 (39.6) | 99 (29.0) | 0.1 |
| Dyslipidemia | 119 (30.2) | 23 (43.4) | 96 (28.2) | **0.02** |
| Heart failure | 64 (16.2) | 14 (26.4) | 50 (14.7) | **0.03** |
| STEMI | 148 (37.6) | 26 (49.1) | 122 (35.8) | 0.06 |
| Shock | 24 (6.1) | 12 (22.6) | 12 (3.5) | **< 0.001** |
| Sepsis | 149 (37.8) | 35 (66.0) | 114 (33.4) | **< 0.001** |
| Creatinine (µmol/L) | 91.6 (77.9–114.4) | 105.4 (84.9–157.7) | 89.9 (77–107.4) | **0.02** |
| Hemoglobin (g/dL) | 12.7 ± 2.2 | 11.8 ± 2.4 | 12.9 ± 2.1 | **0.001** |
| Elevated troponin T | 367 (93.1) | 52 (98.1) | 315 (92.4) | 0.1 |
| NT-proBNP (pg/mL) | 1975 (698–6164) | 6931 (3359–24811) | 1769 (627–4419) | **< 0.001** |
| Ejection fraction (%) | 48 (39–59) | 42 (33–52) | 49 (40–60) | **0.003** |
| Revascularization | 219 (55.6) | 22 (41.5) | 197 (57.8) | **0.02** |
| Heparin | 253 (64.2) | 25 (47.2) | 228 (66.9) | **0.005** |
| Antiplatelets | 355 (90.1) | 35 (66.0) | 320 (93.8) | **< 0.001** |
| Statins | 337 (85.5) | 33 (62.3) | 304 (89.1) | **< 0.001** |
| BB/ACEi/ARB/MRA | 275 (69.8) | 11 (20.8) | 264 (77.4) | **< 0.001** |
| SGLT2i | 94 (23.9) | 3 (5.7) | 91 (26.7) | **< 0.001** |
| Diabetes | 27 (28.7) | 0 | 27 (29.7) | |
| Non-diabetes | 67 (71.3) | 3 (100) | 64 (70.3) | |

STEMI: ST elevation myocardial infarction, NT-proBNP: N-terminal pro B-type natriuretic peptide, BB: beta blockers, ACEi: angiotensin–converting enzyme inhibitors, ARB: angiotensin receptor blockers, MRA: mineralocorticoid antagonists, SGLT2i: sodium–glucose cotransporter-2 inhibitors.

The diagnosis of AMI was based on medical records and verified according to the Fourth Universal Definition of Myocardial Infarction (2018) [5]. Therapeutic management was implemented following the current treatment guidelines of the Viet Nam Ministry of Health [6]. Diagnostic tests and administered medications, including SGLT2i, were recorded from the electronic hospital database, with the time of prescription calculated from the point of hospital admission. Patient prognosis was assessed based on in-hospital survival from the medical records.

Nguyen Trai Hospital is a public tertiary referral center in Ho Chi Minh City, Vietnam. The diagnosis and management of AMI follow the clinical guidelines issued by the Ministry of Health. Laboratory assessments, including troponin T and NT-proBNP, are conducted in the hospital's ISO 15189–accredited laboratory.

All descriptive data are presented as mean ± SD (standard deviation), median, or frequency (%). Continuous variables between the two groups were compared using either the unpaired t-test or the Mann-Whitney U-test, as appropriate. Frequencies were compared using the Chi-squared test. The predictive potential was evaluated using univariable and multivariable logistic regression analyses. Multivariable logistic regression analysis included confounders that were significant in the univariable model. Both univariable and multivariable logistic regression analyses investigated 1-SD changes in continuous variables. NT-proBNP values were natural log–transformed prior to regression analyses. All analyses were performed using SPSS software, version 27.0.

The investigation complied with the principles outlined in the 1975 Declaration of Helsinki. All data were fully anonymized prior to analysis, with no identifiable personal information accessible to the investigators. The study protocol received approval from the institutional review board of Nguyen Trai Hospital (No. 1162/QĐ-BVNT, July 24, 2024), which waived the requirement for informed consent due to the retrospective nature of the study and the use of de-identified data.

## Results

### Study population characteristics

The study population had a relatively high mean age of 66.0 ± 11.7 years, with a tendency toward older age in the event group compared with the non-event group. Nearly two-thirds of participants were male. SGLT2i was initiated within 24 hours post-admission in 23.9% of patients. In-hospital mortality occurred in 53 patients (13.5%) (Table 1).

Left ventricular ejection fraction, shock, sepsis, creatinine, and hemoglobin were variables significantly different between the two groups with and without in-hospital death. NT-proBNP level was markedly higher in patients with events (6931 vs. 1769 pg/mL, p < 0.001). The rates of coronary revascularization, SGLT2i, BB/ACEi/ARB/MRA, antiplatelets, heparin, and statins were significantly higher in the non-event group than in the mortality group (p < 0.05) (Table 1). Left ventricular ejection fraction was lower while BB/ACEi/ARB/MRA use was significantly higher in the SGLT2i group (S1 Table).

### Clinical characteristics associated with in-hospital mortality

Univariable logistic regression analysis of clinical characteristics showed that age, left ventricular ejection fraction, shock, sepsis, serum creatinine, hemoglobin, and NT-proBNP were all significantly associated with in-hospital mortality. However, in the multivariable analysis, only left ventricular ejection fraction (OR 0.91, 95% CI: 0.85–0.97) and sepsis (OR 5.14, 95% CI: 1.04–25.36) remained independently associated with in-hospital mortality (Table 2).

Treatment factors associated with in-hospital outcomes included the use of BB/ACEi/ARB/MRA (OR 0.12, 95% CI: 0.05–0.25) and SGLT2i (OR 0.27, 95% CI: 0.07–0.96) (Table 3). In subgroup univariable analyses, both SGLT2i and heparin were associated with lower in-hospital mortality only among patients who did not undergo revascularization (S2 Table).

After jointly adjusting for clinical characteristics and treatment variables, early SGLT2i use was consistently associated with favorable in-hospital prognosis across the different models evaluated (S3 Table).

**Table 2. Univariable and multivariable logistic regression analyses of clinical characteristics associated with in-hospital mortality.**

| | Univariable | | Multivariable | |
|---|---|---|---|---|
| | OR (95% CI) | P-value | OR (95% CI) | P-value |
| Age (years) | 1.05 (1.02–1.08) | **< 0.001** | 1.05 (0.97–1.12) | 0.2 |
| Male gender | 0.75 (0.42–1.34) | 0.3 | --- | --- |
| Shock | 8.02 (3.38–19.03) | **< 0.001** | 2.81 (0.44–17.81) | 0.3 |
| Sepsis | 3.87 (2.10–7.13) | **< 0.001** | 5.14 (1.04–25.36) | **0.04** |
| Creatinine (µmol/L) | 1.02 (1.01–1.04) | **0.002** | 1.02 (0.99–1.04) | 0.05 |
| Hemoglobin (g/dL) | 0.79 (0.69–0.92) | **0.002** | 0.92 (0.63–1.35) | 0.6 |
| NT-proBNP (log-transformed) | 1.85 (1.48–2.30) | **< 0.001** | 1.56 (0.93–2.64) | 0.09 |
| Elevated troponin T | 4.29 (0.57–32.31) | 0.1 | --- | --- |
| Ejection fraction (%) | 0.97 (0.95–0.98) | **0.004** | 0.91 (0.85–0.97) | **0.003** |

NT-proBNP: N-terminal pro B-type natriuretic peptide.

**Table 3. Univariable and multivariable logistic regression analyses of treatment factors associated with in-hospital outcomes.**

| | Univariable | | Multivariable | |
| --- | --- | --- | --- | --- |
| | OR (95% CI) | P-value | OR (95% CI) | P-value |
| Revascularization | 0.52 (0.29–0.93) | **0.02** | 0.58 (0.29–1.11) | 0.1 |
| Heparin | 0.44 (0.25–0.79) | **0.006** | 0.79 (0.38–1.63) | 0.5 |
| Antiplatelets | 0.13 (0.06–0.26) | **< 0.001** | 0.37 (0.12–1.14) | 0.08 |
| Statins | 0.20 (0.10–0.38) | **< 0.001** | 0.69 (0.25–1.97) | 0.4 |
| BB/ACEi/ARB/MRA | 0.08 (0.04–0.15) | **< 0.001** | 0.12 (0.05–0.25) | **< 0.001** |
| SGLT2i | 0.16 (0.05–0.54) | **0.003** | 0.27 (0.07–0.96) | **0.04** |

BB: beta blockers, ACEi: angiotensin–converting enzyme inhibitors, ARB: angiotensin receptor blockers, MRA: mineralocorticoid antagonists, SGLT2i: sodium–glucose cotransporter-2 inhibitors.

## Discussion

The analysis of 394 patients with AMI admitted to a tertiary hospital in Ho Chi Minh City, Vietnam, demonstrated that initiation of SGLT2i within 24 hours of admission was independently associated with lower in-hospital mortality. Additionally, reduced left ventricular ejection fraction and the presence of sepsis were associated with adverse prognosis, whereas guideline-directed medical therapy was linked to improved in-hospital survival.

Among comorbidities, diabetes mellitus was present in 30.5% of participants, with a higher proportion in the mortality group, although the difference was not statistically significant. Heart failure occurred more frequently in patients with adverse outcomes (26.4% vs. 14.7%, p = 0.03). Due to the retrospective nature of the study, it was not possible to distinguish whether heart failure was pre-existing or de novo in association with AMI. Large randomized controlled trials, such as DAPA-MI and EMPACT-MI, did not show statistically significant differences in heart failure hospitalization or all-cause mortality after AMI between the SGLT2i and control groups [7,8]. This difference may be explained by the later initiation of SGLT2i (mean 5–10 days) in the two aforementioned trials compared with our study (within 24 hours). Delayed initiation may miss the "window period" after myocardial infarction—a phase during which left ventricular injury and remodeling progress rapidly and are particularly sensitive to the effects of SGLT2 inhibition. Findings from a secondary analysis of the EMMY trial demonstrated that initiating empagliflozin within 24 hours after percutaneous coronary intervention resulted in a reduction in NT-proBNP comparable to that achieved with later initiation (24–72 hours), while also improving left ventricular functional and structural parameters without increasing the incidence of serious adverse events [9]. The exclusion of participants with common high-risk features in the AMI population, such as diabetes mellitus and heart failure, in the DAPA-MI trial may have reduced the ability to detect the true therapeutic benefit of the drug. A recent meta-analysis identified diabetes as a key effect modifier, demonstrating that for every 1% increase in the prevalence of diabetes in the study population, the all-cause mortality rate decreased by 0.49% [10]. This indicates that patients with diabetes mellitus are particularly sensitive to the cardiovascular benefits of SGLT2i, especially when treatment is initiated early.

It is noteworthy that a considerable proportion of patients in this study did not undergo coronary revascularization, a common scenario in developing countries, and represents a notable difference compared with strictly controlled clinical trials. Indeed, our subgroup analyses showed that the association of early SGLT2i use with in-hospital outcome was observed only in patients who did not undergo coronary revascularization. Failure to achieve coronary revascularization leads to a larger ischemic burden, elevated left ventricular filling pressures, and accelerated adverse ventricular remodeling, rapidly predisposing patients to acute heart failure and early mortality. In this high-risk setting, the favorable clinical associations observed with early SGLT2i use are consistent with rapid hemodynamic, metabolic, and cellular effects. SGLT2 inhibition improves ventricular loading conditions through osmotic diuresis and natriuresis, preferentially reducing

interstitial congestion, lowering preload, and improving arterial stiffness and afterload without provoking excessive neurohormonal activation. At the same time, restoration of tubuloglomerular feedback lowers intraglomerular pressure and mitigates acute kidney injury, thereby stabilizing the cardiorenal axis during the vulnerable post-infarction phase [11,12]. Beyond hemodynamics, SGLT2 inhibition induces favorable myocardial metabolic remodeling by shifting substrate utilization away from inefficient glucose metabolism toward ketone bodies and fatty acid oxidation, thereby enhancing myocardial ATP production and mechanical efficiency. Experimental models of post-AMI heart failure demonstrate that empagliflozin reduces LV dilatation, improves systolic function, and attenuates adverse remodeling despite similar infarct size at baseline [13]. At the cellular level, SGLT2 inhibition activates a fasting-like transcriptional program via AMPK and SIRT1 while suppressing Akt/mTOR signaling, promoting autophagy, reducing oxidative stress and inflammation, and preserving mitochondrial function in both the heart and kidney [14]. These coordinated effects provide a plausible biological explanation for the early favorable clinical associations observed in patients with AMI, particularly among those who do not undergo coronary revascularization.

This study was conducted at a high-volume tertiary cardiovascular center that admits a substantial number of patients with AMI, thereby partially reflecting the real-world diagnosis and management practices in tertiary hospitals in Ho Chi Minh City. Data were obtained from electronic medical records with high completeness and detail, ensuring reliability in the analysis of both clinical characteristics and in-hospital treatment. However, several limitations should be acknowledged. As the study was conducted at a single center, its generalizability to the overall population of patients with AMI in Vietnam may be limited. Additionally, the retrospective design carries inherent constraints in data collection, particularly regarding the rationale for early SGLT2i initiation (whether related to diabetes mellitus, heart failure, or chronic kidney disease). Thus, the decision to initiate SGLT2i might not be random and influenced by baseline heart failure severity, as reflected by the lower LVEF in the SGLT2i group. These limitations underscore the need for larger, multicenter, prospective cohort studies to further elucidate the prognostic factors and therapeutic strategies, including early SGLT2 inhibition, that may be associated with improved outcomes in AMI within the Vietnamese healthcare setting.

## Conclusions

Early initiation of SGLT2i within 24 hours of hospital admission was independently associated with lower in-hospital mortality in patients with AMI. This association was most pronounced in high-risk patients, particularly those who did not undergo coronary revascularization. These findings suggest a potential association between early SGLT2i use and in-hospital outcomes, which warrants confirmation in prospective randomized trials.

## Supporting information

**S1 Table. Comparison of baseline characteristics between the SGLT2i and non-SGLT2i groups.**
(DOCX)

**S2 Table. Univariable analysis of clinical and treatment characteristics associated with in-hospital outcomes between the non-revascularization and revascularization groups.**
(DOCX)

**S3 Table. Association between SGLT2i use and in-hospital mortality across multiple univariable and multivariable logistic regression models.**
(DOCX)

## Acknowledgments

The authors gratefully acknowledge Minh Van Le and Thien Ngoc Nguyen for their assistance in sample collection.

## Author contributions

**Conceptualization:** Si Van Nguyen.

**Data curation:** Anh Thi Kim Nguyen, Tuan Minh Nguyen.

**Formal analysis:** Bao Quoc Dinh.

**Funding acquisition:** Hung Thanh Quach.

**Investigation:** Anh Thi Kim Nguyen, Tuan Minh Nguyen, Si Van Nguyen.

**Methodology:** An Le Pham, Si Van Nguyen.

**Project administration:** Hung Thanh Quach, Si Van Nguyen.

**Resources:** Hung Thanh Quach, Anh Thi Kim Nguyen, Tuan Ngoc Tran, Tien Dung Ho.

**Software:** Bao Quoc Dinh.

**Supervision:** Hung Thanh Quach, Hoang Huy Bui, Tuan Ngoc Tran, Tien Dung Ho, An Le Pham, Si Van Nguyen.

**Validation:** Hung Thanh Quach, Bao Quoc Dinh, Si Van Nguyen.

**Visualization:** Bao Quoc Dinh.

**Writing – original draft:** Si Van Nguyen.

**Writing – review & editing:** Hung Thanh Quach, Anh Thi Kim Nguyen, Tuan Minh Nguyen, Hoang Huy Bui, Tuan Ngoc Tran, Tien Dung Ho, An Le Pham, Si Van Nguyen.

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
