## [Decision Letter · Decision Letter 0]

4 Dec 2025

Dear Dr. Nguyen,

Thank you for submitting your manuscript to PLOS ONE. After careful consideration, we feel that it has merit but does not fully meet PLOS ONE’s publication criteria as it currently stands. Therefore, we invite you to submit a revised version of the manuscript that addresses the points raised during the review process.

We look forward to receiving your revised manuscript.

Kind regards,

Satoshi Higuchi

Academic Editor

PLOS ONE

Journal Requirements:

4. Please include captions for your Supporting Information files at the end of your manuscript, and update any in-text citations to match accordingly. Please see our Supporting Information guidelines for more information: http://journals.plos.org/plosone/s/supporting-information ..

Reviewers' comments:

Reviewer's Responses to Questions

**Comments to the Author**

1. Is the manuscript technically sound, and do the data support the conclusions?

Reviewer #1: Partly

Reviewer #2: Yes

2. Has the statistical analysis been performed appropriately and rigorously?

Reviewer #1: Yes

Reviewer #2: Yes

3. Have the authors made all data underlying the findings in their manuscript fully available?

Reviewer #1: No

Reviewer #2: Yes

4. Is the manuscript presented in an intelligible fashion and written in standard English?

Reviewer #1: Yes

Reviewer #2: Yes

Reviewer #1: Study Design and Generalizability: The authors correctly identify the study's single-center, retrospective design as a key limitation. This design introduces the possibility of selection bias, as the decision to initiate SGLT2i was not random. It's not clear from the manuscript whether the patients who received early SGLT2i were fundamentally different from those who did not, beyond the documented clinical characteristics. This could affect the generalizability of the findings.

Patient Population: The study notes that a "considerable proportion" of patients did not undergo coronary revascularization. This is a significant point of difference compared to many clinical trials and might contribute to the observed benefits of SGLT2i in this specific, high-risk population. While the authors mention this in the discussion, a deeper analysis or discussion of the impact of this factor on the results would strengthen the paper.

Statistical Analysis: The multivariate logistic regression model identified a few key factors independently associated with in-hospital mortality, including left ventricular ejection fraction, sepsis, BB/ACEi/ARB/MRA use, and early SGLT2i initiation. The odds ratio (OR) for BB/ACEi/ARB/MRA use is 0.1 (95% CI: 0-0.2). An odds ratio with a lower bound of 0 suggests a very strong protective effect, and the confidence interval is quite wide. The authors should consider providing more context on how this very low OR was interpreted.

Clarity in Abstract: The abstract mentions that the study was conducted from January 2022 to October 2024. The methods section also mentions data assessment from July 31 to October 24, 2024. Clarifying the purpose of the two different date ranges would improve clarity.

Reviewer #2: Reviewer Comments

I would like to congratulate the authors for this well-designed and clinically relevant study. Please consider the following revisions:

Major Revisions:

Tables 2 and 3 show separate multivariable models, but it is unclear whether a single combined model including both clinical and treatment variables was tested. Because LVEF, sepsis, renal function, and revascularization strongly influence prognosis and treatment selection, a unified model is essential to determine whether early SGLT2i initiation remains independently associated with in-hospital mortality. If not performed, I strongly encourage including such a model—at least as a sensitivity analysis—to minimize residual confounding.

Discussion:

The Discussion would benefit from a more comprehensive description of the mechanisms that may explain the early benefits of SGLT2 inhibitors in acute MI. Please expand on hemodynamic effects (osmotic diuresis, natriuresis), improvements in vascular function, renal protection, anti-inflammatory and anti-oxidative actions, and especially metabolic remodeling involving increased ketone-based myocardial fuel utilization. To support this, consider citing key mechanistic reviews (Ferrannini 2014; Verma & McMurray 2018; Zelniker & Braunwald 2018; Packer 2020; Santos-Gallego 2019). Strengthening this section will enhance the mechanistic plausibility of your findings.

Minor Revisions:

The manuscript would benefit from a more detailed characterization of patients with diabetes. Although prevalence is similar between groups, several key aspects remain unclear. Please report baseline HbA1c levels to assess metabolic control, and, if available, information on diabetes-related complications (nephropathy, neuropathy, retinopathy, macrovascular disease) to better describe disease severity. It would also be useful to clarify whether the myocardial infarction was a first event or a recurrence in patients with diabetes. Given the relevance of SGLT2 inhibitors in type 2 diabetes, I suggest performing subgroup analyses to evaluate whether the association between early SGLT2i initiation and outcomes varies by HbA1c categories, CKD severity, or the presence of diabetes complications.

**Do you want your identity to be public for this peer review?** For information about this choice, including consent withdrawal, please see our For information about this choice, including consent withdrawal, please see our Privacy Policy .

Reviewer #1: **Yes:** Gerrit JacobGerrit Jacob

Reviewer #2: No

---

## [Author Response · Author response to Decision Letter 1]

12 Jan 2026

Reviewer 1

1. Study Design and Generalizability: The authors correctly identify the study's single-center, retrospective design as a key limitation. This design introduces the possibility of selection bias, as the decision to initiate SGLT2i was not random. It's not clear from the manuscript whether the patients who received early SGLT2i were fundamentally different from those who did not, beyond the documented clinical characteristics. This could affect the generalizability of the findings.

Thank you very much for this important comment. Because of the retrospective nature of the study, the decision to initiate SGLT2i was not random. We compared the SGLT2i and non-SGLT2i groups and found that only LVEF was significantly lower in the SGLT2i group (Table S1). However, it was not possible to determine whether heart failure was pre-existing or occurred as a consequence of acute myocardial infarction. In patients with pre-existing heart failure, which is a strong adverse prognostic factor, SGLT2i has been shown to provide clinical benefit. Therefore, although non-random treatment selection may have occurred, this imbalance may also reflect preferential use of SGLT2i in higher-risk patients.

We added this information to the Results section as:

“LVEF was lower while BB/ACEi/ARB/MRA use was significantly higher in the SGLT2i group (Table S1)” (page 5, first paragraph, last two lines in red font), and to the Discussion section as:

“Thus, the decision to initiate SGLT2i might not have been random and influenced by baseline heart failure severity, as reflected by the lower LVEF in the SGLT2i group” (page 9, first paragraph, lines 2–4 in red font), as well as in Table S1 in the Supplementary Data.

2. Patient Population: The study notes that a "considerable proportion" of patients did not undergo coronary revascularization. This is a significant point of difference compared to many clinical trials and might contribute to the observed benefits of SGLT2i in this specific, high-risk population. While the authors mention this in the discussion, a deeper analysis or discussion of the impact of this factor on the results would strengthen the paper.

Thank you for this important suggestion. We performed subgroup univariable regression analyses to evaluate associations between clinical and treatment-related factors and in-hospital mortality in patients with and without coronary revascularization. SGLT2i use was statistically significant only in the non-revascularized group.

Accordingly, we added these findings to the Results and Discussion sections as:

“In subgroup univariate analyses, SGLT2i together with heparin was only associated with improved in-hospital mortality among patients who did not undergo revascularization (Table S2)” (page 6, paragraph 2, last three lines in red font), and

“Indeed, our subgroup analyses showed that the association between early SGLT2i use and in-hospital outcome was observed only in patients who did not undergo coronary revascularization” (page 8, paragraph 1, lines 3-5 in red font), as well as in Table S2 of the Supplementary Data.

3. Statistical Analysis: The multivariate logistic regression model identified a few key factors independently associated with in-hospital mortality, including left ventricular ejection fraction, sepsis, BB/ACEi/ARB/MRA use, and early SGLT2i initiation. The odds ratio (OR) for BB/ACEi/ARB/MRA use is 0.1 (95% CI: 0-0.2). An odds ratio with a lower bound of 0 suggests a very strong protective effect, and the confidence interval is quite wide. The authors should consider providing more context on how this very low OR was interpreted.

Thank you for this valuable comment. Because of the limited number of events, we used a composite variable including medications that affect the renin–angiotensin–aldosterone and sympathetic nervous systems. Each of these drug classes has well-established benefits in acute myocardial infarction. Therefore, their combined effect may explain the very low odds ratio observed.

4. Clarity in Abstract: The abstract mentions that the study was conducted from January 2022 to October 2024. The methods section also mentions data assessment from July 31 to October 24, 2024. Clarifying the purpose of the two different date ranges would improve clarity.

We apologize for the lack of clarity. The period from January 2022 to October 2024 refers to the study period during which patients were hospitalized and treated. The period from July 31 to October 24, 2024 corresponds to the timeframe during which electronic medical records were retrospectively accessed and analyzed.

We revised the Abstract (Methods) as follows:

“A retrospective, single-center study was conducted on 394 adult patients hospitalized with acute myocardial infarction at Nguyen Trai Hospital, Ho Chi Minh City, between January 2022 and October 2024. The data extraction and analysis were performed from July 31 to October 24, 2024.”

Reviewer 2

1. Tables 2 and 3 show separate multivariable models, but it is unclear whether a single combined model including both clinical and treatment variables was tested. Because LVEF, sepsis, renal function, and revascularization strongly influence prognosis and treatment selection, a unified model is essential to determine whether early SGLT2i initiation remains independently associated with in-hospital mortality. If not performed, I strongly encourage including such a model—at least as a sensitivity analysis—to minimize residual confounding.

Thank you very much for this important suggestion. We analyzed prognostic factors, including SGLT2i use, across three regression models that incorporated both clinical characteristics and treatment variables. SGLT2i remained associated with in-hospital mortality across these models. These results are reported in the Results section as:

“After jointly adjusting for clinical characteristics and treatment variables, early SGLT2i use was consistently associated with favorable in-hospital prognosis across the different models evaluated (Table S3)” (page 6, paragraph 2 in red font) and in Table S3 of the Supplementary Data.

2. The Discussion would benefit from a more comprehensive description of the mechanisms that may explain the early benefits of SGLT2 inhibitors in acute MI. Please expand on hemodynamic effects (osmotic diuresis, natriuresis), improvements in vascular function, renal protection, anti-inflammatory and anti-oxidative actions, and especially metabolic remodeling involving increased ketone-based myocardial fuel utilization. To support this, consider citing key mechanistic reviews (Ferrannini 2014; Verma & McMurray 2018; Zelniker & Braunwald 2018; Packer 2020; Santos-Gallego 2019). Strengthening this section will enhance the mechanistic plausibility of your findings.

Thank you for your valuable suggestion and the recommended references. We expanded the mechanistic discussion to include hemodynamic, metabolic, and cellular effects of SGLT2i in paragraph 3 of the Discussion (page 7, last paragraph and page 8, first paragraph), with supporting references 11–14.

---

## [Decision Letter · Decision Letter 1]

26 Feb 2026

Dear Dr. Nguyen,

Thank you for submitting your manuscript to PLOS ONE. After careful consideration, we feel that it has merit but does not fully meet PLOS ONE’s publication criteria as it currently stands. Therefore, we invite you to submit a revised version of the manuscript that addresses the points raised during the review process.

We look forward to receiving your revised manuscript.

Kind regards,

Satoshi Higuchi

Academic Editor

PLOS One

**Journal Requirements:**

**Additional Editor Comments:**

Thank you for submitting your revised manuscript.

Your work is close to acceptance.

However, I would ask the authors to avoid using the term “effect,” as this is an observational study and therefore cannot establish a causal relationship.

For example, a title such as “Favorable association between early initiation of sodium-glucose cotransporter-2 inhibitors and in-hospital prognosis in acute myocardial infarction” would be preferable.

Please revise the manuscript accordingly and ensure that causal language is avoided throughout the manuscript.

Reviewers' comments:

Reviewer's Responses to Questions

**Comments to the Author**

Reviewer #2: All comments have been addressed

2. Is the manuscript technically sound, and do the data support the conclusions?

Reviewer #2: Yes

3. Has the statistical analysis been performed appropriately and rigorously?

Reviewer #2: Yes

4. Have the authors made all data underlying the findings in their manuscript fully available?

Reviewer #2: Yes

5. Is the manuscript presented in an intelligible fashion and written in standard English?

Reviewer #2: Yes

**Reviewer #2:** I would like to thank the authors for their thorough and thoughtful revision. They have adequately addressed all the reviewers’ comments, substantially strengthening the manuscript, particularly with respect to the multivariable analyses, subgroup analyses, and the expanded mechanistic discussion. The methodology is now clearer, the results are robustly supported, and the discussion is well balanced and clinically relevant.I would like to thank the authors for their thorough and thoughtful revision. They have adequately addressed all the reviewers’ comments, substantially strengthening the manuscript, particularly with respect to the multivariable analyses, subgroup analyses, and the expanded mechanistic discussion. The methodology is now clearer, the results are robustly supported, and the discussion is well balanced and clinically relevant.

In its current form, I believe the manuscript is suitable for publication.

**Do you want your identity to be public for this peer review?** For information about this choice, including consent withdrawal, please see our For information about this choice, including consent withdrawal, please see our Privacy Policy .

Reviewer #2: **Yes:** Pedro Gil-MillánPedro Gil-Millán

---

## [Author Response · Author response to Decision Letter 2]

28 Feb 2026

Additional Editor Comments:

Thank you for submitting your revised manuscript.

Your work is close to acceptance.

However, I would ask the authors to avoid using the term “effect,” as this is an observational study and therefore cannot establish a causal relationship.

For example, a title such as “Favorable association between early initiation of sodium-glucose cotransporter-2 inhibitors and in-hospital prognosis in acute myocardial infarction” would be preferable.

Please revise the manuscript accordingly and ensure that causal language is avoided throughout the manuscript.

 We sincerely thank the Academic Editor for this important and constructive comment. In response, we have revised the title to: “Favorable Association Between Early Initiation of Sodium-Glucose Cotransporter-2 Inhibitors and In-Hospital Prognosis in Acute Myocardial Infarction.”

We have also carefully reviewed the entire manuscript and replaced causal terminology with association-neutral language where appropriate. The abstract, results, discussion, and conclusion sections were revised to emphasize that this observational study identifies associations rather than causal relationships. We further clarified that the findings are hypothesis-generating and warrant confirmation in prospective randomized trials.

All modifications are highlighted in red in the revised manuscript.

Reviewer #2:

I would like to thank the authors for their thorough and thoughtful revision. They have adequately addressed all the reviewers’ comments, substantially strengthening the manuscript, particularly with respect to the multivariable analyses, subgroup analyses, and the expanded mechanistic discussion. The methodology is now clearer, the results are robustly supported, and the discussion is well balanced and clinically relevant.

In its current form, I believe the manuscript is suitable for publication.

 We sincerely thank the reviewer for the thoughtful evaluation and supportive comments. We greatly appreciate the recognition of the strengthened multivariable analyses, subgroup analyses, and expanded mechanistic discussion.

To improve numerical precision and clarity, odds ratios are now reported with two decimal places instead of one. In addition, NT-proBNP values were natural log–transformed for regression analyses to enhance model stability and interpretability. These refinements did not alter the statistical significance or the overall interpretation of the findings.

All related changes are highlighted in blue in the revised manuscript.

---

## [Editor Report · Decision Letter 2]

4 Mar 2026

FAVORABLE ASSOCIATION BETWEEN EARLY INITIATION OF SODIUM-GLUCOSE COTRANSPORTER-2 INHIBITORS AND IN-HOSPITAL PROGNOSIS IN ACUTE MYOCARDIAL INFARCTION

PONE-D-25-44149R2

Dear Dr. Nguyen,

We’re pleased to inform you that your manuscript has been judged scientifically suitable for publication and will be formally accepted for publication once it meets all outstanding technical requirements.

Kind regards,

Satoshi Higuchi

Academic Editor

PLOS One

Additional Editor Comments (optional):

Thank you for your careful revision and for submitting this well-written manuscript.

---

## [Editor Report · Acceptance letter]

PONE-D-25-44149R2

PLOS One

Dear Dr. Nguyen,

I'm pleased to inform you that your manuscript has been deemed suitable for publication in PLOS One. Congratulations! Your manuscript is now being handed over to our production team.

Kind regards,

on behalf of

Dr. Satoshi Higuchi

Academic Editor

PLOS One